# An optimized transformer model for efficient detection of thoracic diseases in chest X-rays with multi-scale feature fusion

Shasha Yu[1], Peng Zhou[2]*

1 Information Center, Zhongnan Hospital of Wuhan University, Wuhan, China, 2 FutureFront Interdisciplinary Research Institute, Huazhong University of Science and Technology, Wuhan, China

* peng_zhou@hust.edu.cn

## Abstract

This study presents the development and application of an optimized Detection Transformer (DETR) model, known as CD-DETR, for the detection of thoracic diseases from chest X-ray (CXR) images. The CD-DETR model addresses the challenges of detecting minor pathologies in CXRs, particularly in regions with uneven medical resource distribution. In the central and western regions of China, due to a shortage of radiologists, CXRs from township hospitals are concentrated in central hospitals for diagnosis. This requires processing a large number of CXRs in a short period of time to obtain results. The model integrates a multi-scale feature fusion approach, leveraging Efficient Channel Attention (ECA-Net) and Spatial Attention Upsampling (SAU) to enhance feature representation and improve detection accuracy. It also introduces a dedicated Chest Diseases Intersection over Union (CDIoU) loss function to optimize the detection of small targets and reduce class imbalance. Experimental results on the NIH Chest X-ray dataset demonstrate that CD-DETR achieves a precision of 88.3% and recall of 86.6%, outperforming other DETR variants by an average of 5% and CNN-based models like YOLOv7 by 6–8% in these metrics, showing its potential for practical application in medical imaging diagnostics.

## 1. Introduction

Chest X-ray (CXR) is a widely used imaging technique for observing structures within the thoracic cavity, including the heart, lungs, and chest wall. It plays a significant role in diagnosing various thoracic diseases, such as pulmonary diseases, heart diseases, pleural and pleural cavity diseases, etc. Due to its low cost, ease of operation, and low radiation dose, CXR is often the preferred method for initial screening and diagnosis of many thoracic diseases [1]. However, due to the disparity in medical resources in China, the examination efficiency of CXR technology is extremely low in many parts of the country [2]. There is a significant geographical disparity in medical resources in

**Data availability statement:** NIH Chest X-ray dataset is available from https://www.kaggle.com/datasets/nih-chest-xrays/data; VinBigData dataset is available from https://kaggle.com/competitions/vinbigdata-chest-xray-abnormalities-detection.

**Funding:** The author(s) received no specific funding for this work.

**Competing interests:** The authors have declared that no competing interests exist.

China, with the eastern region having more tertiary hospitals compared to the central and western regions. Typically, within China's medical system, tertiary hospitals are the highest level, usually equipped with advanced medical equipment, abundant medical resources, and high-level medical services. This imbalance in medical resources is even more pronounced between urban and rural areas. Remote areas have relatively lower medical standards and a shortage of medical facilities and physician resources. Almost all of the top 100 hospitals in China are located in major cities or provincial capitals in the southeastern region. The distribution of radiologists in different regions of China is even more apparent. The eastern region has the highest number of radiologists, while the western region has relatively fewer. The number of radiologists in the eastern region is almost three times that of the western region, leading to differences in the accessibility and quality of medical services [3].

To address the shortage of radiologists in central and western China, particularly in rural areas, Chinese medical institutions have adopted a centralized image reading model. Centralized image reading involves a daily collective review session at a designated time led by a department head or senior physician, with all radiologists participating to review medical imaging data uploaded from rural hospitals. This model has alleviated the shortage of radiologists in township areas to some extent. However, it significantly increases the workload of doctors at the image reading center. Especially during large-scale outbreaks of lung diseases, a backlog of CXR awaiting diagnosis can accumulate at the center, and delayed processing poses risks of diagnostic delays and disease spread. Additionally, the high-intensity workload can lead to errors, including missed and incorrect diagnoses [4]. Thus, many regions in China have established an AI-assisted centralized image reading method, as shown in Fig 1. First, township hospitals upload collected CXR images to the central hospital's medical imaging database. Then, an AI model conducts an intelligent screening, identifying abnormal or suspected cases, which are then forwarded to specialist doctors for diagnosis. Cases without abnormalities are placed in a normal database for random doctor review to ensure no oversights. In this model, radiologists only need to focus on CXRs flagged by the AI model as potentially diseased. This human-machine collaboration significantly reduces the workload of radiologists and greatly improves the efficiency of the image reading center.

The centralized image reading model enhances the efficiency of image reading centers and helps address the imbalance in radiologist resources to some extent. However, this model presents challenges for the AI model. First, the AI model must ensure thorough screening without any missed cases. The mechanism of random doctor reviews is primarily a safeguard against large-scale unknown diseases but does not fundamentally eliminate the risk of missed diagnoses. Secondly, early disease detection is critical. Early-stage diseases, due to subtle symptoms, may have indistinct imaging signs, so the AI model should be capable of identifying these early conditions. Finally, in the event of a sudden outbreak of a widespread infectious disease, such as COVID-19, a large volume of images from township hospitals will accumulate in the medical imaging database, requiring rapid processing. This demands that the AI model handle these cases at high speed, posing a challenge

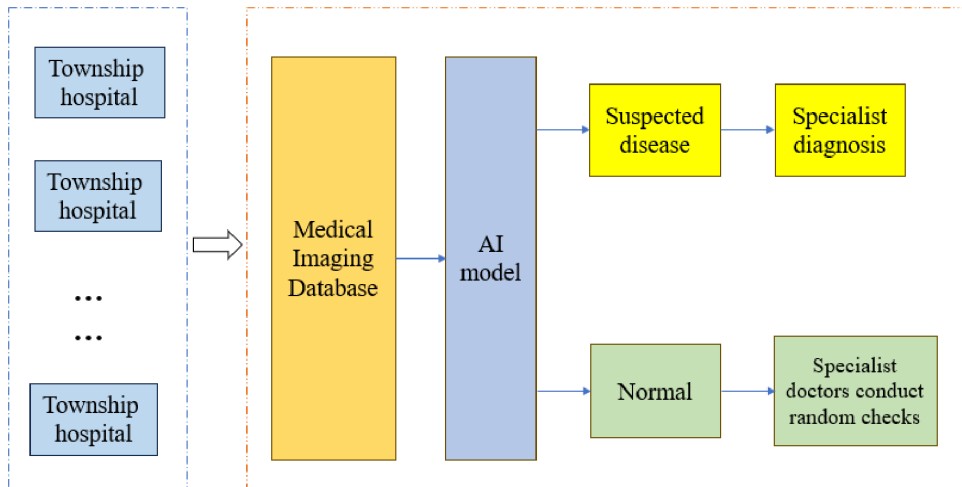

**Fig 1. An AI-assisted centralized image reading method.**

to its processing capability. In recent years, the DETR (Detection Transformer) model has been applied in the detection of thoracic diseases from CXR images. DETR is a deep learning model designed for object detection tasks. It transforms the object detection problem into a direct set prediction problem, predicting the categories and bounding boxes of a set of objects simultaneously, rather than predicting each object independently. This approach helps the model learn global contextual information and accurately predict the thoracic diseases learned during training.

Despite DETR's many advantages, it also faces some challenges when applied to CXR detection. As the depth of the network increases, the size of the feature maps progressively decreases, leading to a loss of crucial information, especially for small lesions. This diminishment of feature resolution limits the model's ability to detect minor pathologies, making it less effective at identifying subtle abnormalities in CXR images. The self-attention mechanism in DETR can overwhelm the limited and subtle information in small lesions, causing it to focus more on the larger or more prominent features. This leads to contamination from background features, resulting in the model losing focus on the target lesions. Consequently, the accuracy of detection for small or subtle abnormalities can suffer. CXR images often contain high background noise, such as rib structures, shadows, and air spaces, which can distract the model from focusing on the relevant pathological features. The model's reliance on global feature extraction might cause it to overfit to these irrelevant background elements, leading to potential misdetections or false positives.

These challenges hinder DETR's full potential for precise and efficient CXR detection, especially for small, subtle abnormalities and in low-resource clinical settings. Researchers continue to explore ways to address these issues, such as by enhancing feature resolution, improving attention mechanisms, and integrating more robust training strategies. To address these challenges, we propose the CD-DETR model. The innovations of this model are twofold. First, by combining low-level features from the backbone network with high-level features from the Transformer via the ECA-Net and SAU fusion module, the model enhances its ability to capture contextual information. Second, by solving the instability in the matching process caused by minor symptoms with the CDIoU loss function, it accelerates training speed.

By incorporating these improvements into the DETR variant, we aim to enhance the detection performance of CXR images, especially for minor pathologies. The experimental results demonstrate the effectiveness of the proposed methods and emphasize their potential for real-world applications. These advancements ultimately contribute to the ongoing progress in object detection, facilitating more precise and efficient detection of thoracic diseases.

The main contributions of this study are as follows:

- **Development of CD-DETR**: We propose an optimized Detection Transformer model, CD-DETR, tailored for CXR analysis, addressing the limitations of standard DETR in detecting small and subtle thoracic pathologies.

- **Multi-Scale Feature Fusion**: We introduce a novel fusion strategy combining Efficient Channel Attention (ECA-Net) and Spatial Attention Upsampling (SAU), enhancing the model's ability to capture contextual information across scales.

- **CDIoU Loss Function**: We design a specialized Chest Diseases Intersection over Union (CDIoU) loss function to improve small object detection and mitigate class imbalance, accelerating training stability.

- **Practical Impact**: We demonstrate the model's applicability in AI-assisted centralized image reading systems, improving diagnostic efficiency in resource-scarce regions, with superior performance validated on benchmark datasets.

## 2. Related works

### 2.1 Challenges in CXR image recognition

CXR is a common and important diagnostic tool in medical imaging, widely used to detect and assess the health of the lungs and cardiovascular system [5]. CXR plays a crucial role in diagnosing respiratory and cardiovascular diseases, particularly in the early detection and monitoring of conditions such as pneumonia, tuberculosis, and lung cancer. With the continuous advancement of medical imaging technology, automated CXR analysis is becoming increasingly important in clinical practice and healthcare [6]. By introducing automated analysis, physicians can more efficiently process and interpret a large number of X-ray images, providing more accurate diagnostic support, thereby improving the quality of diagnosis and treatment and work efficiency, especially in resource-constrained areas and environments.

CXR image recognition faces several challenges, one of which is image variability. For example, differences in X-ray machines and patient positioning can lead to variations in image quality and detail [1], thereby affecting the performance of the model. Moreover, subtle abnormalities in CXR images are often difficult to detect, especially as early disease manifestations may not be easily recognized, adding greater difficulty to automated analysis. Another challenge is the limited annotated datasets and class imbalance [7]. In CXR image recognition, particularly in the detection of rare diseases, training datasets often lack sufficient labeled samples, and the imbalance between different classes of targets (such as normal and abnormal categories) may also cause the model to bias towards more common categories, affecting the accuracy and reliability of the model.

### 2.2 Progress of AI in CXR image recognition

Early methods of CXR image recognition primarily relied on rule-based anomaly detection systems, such as template matching [8] and edge detection [9], which used predefined rules to identify abnormal regions in images. With the advancement of technology, early machine learning models were applied to the classification and detection of CXR images [10]. Additionally, handcrafted feature extraction methods, such as Gabor filters and texture analysis, were introduced to extract key features from images, and were combined with machine learning models like support vector machines and random forests for training and prediction [11]. However, these early methods also had limitations: first, they heavily relied on manual feature extraction, which not only increased the complexity of feature selection but also limited the automation capability of the models; secondly, these methods lacked generalizability across different CXR datasets, making it difficult to adapt to images from different devices and patient conditions, leading to unstable performance in practical applications.

With the advancement of CXR image recognition technology, more complex architectures and methods have been widely applied to improve performance and accuracy. Transfer learning, as an effective technique, utilizes pre-trained convolutional neural network (CNN) models on large image datasets (such as ImageNet) to handle CXR tasks. Case

studies such as DenseNet [12], and ResNet [13] have made significant progress in CXR image recognition, with the advantage of improving performance and accelerating model convergence even with limited labeled data. Another important advancement is multi-task learning, which provides a more efficient solution for disease detection in real-world clinical settings by simultaneously detecting multiple diseases such as pneumonia, tuberculosis, and lung nodules. Furthermore, the introduction of attention mechanisms has further improved the performance of CXR image recognition [14], particularly in focusing on relevant regions (e.g., lung fields and nodules), with typical applications including Attention U-Net [15] and Mask R-CNN [16]. Finally, the emergence of end-to-end training methods has reduced the reliance on manual preprocessing steps, enabling direct diagnostic predictions from CXR images, thereby further improving the efficiency and accuracy of automated diagnostic workflows. Recent advancements also include hybrid CNN approaches for pneumonia detection [17,18] and deep learning models for lung disease classification [19], demonstrating the growing role of AI in thoracic diagnostics.

The YOLO series of models have been widely used in CNN-based object detection models in recent years. Tsai [20] used the cascading YOLO5 for rib fracture detection in frontal and oblique CXR images, which can reliably determine the location of cracks. Fan [21] applied the YOLO5 algorithm to the CXR anomaly target detection method, with accuracies 7.28% and 5.89% higher than the Faster RCNN [22] algorithm and EfficientDet [23] algorithm, respectively. Bista applied the model based on YOLO7 to the detection of tuberculosis in CXR examination, and achieved good results. Although YOLO has made certain progress in small object detection through multiple optimizations, tiny lung nodules or other subtle lesions in CXR images are still difficult to detect precisely. YOLO may not be able to fully capture these small objects, especially when the objects are small and similar to the background.

## 2.3 DETR in CXR image recognition

DETR is a target detection framework based on the Transformer architecture, which innovatively transforms the target detection problem into a sequence-to-sequence task, no longer relying on traditional candidate box generation methods [24]. The application of DETR technology in CXR image recognition has demonstrated its advantages.

Firstly, DETR achieves end-to-end target detection through the Transformer architecture. Unlike YOLO [25], which requires manual design of candidate boxes, DETR directly extracts global features from CXR images using self-attention mechanisms. This global modeling capability enables DETR to effectively capture lesion areas in complex and blurred CXR images, especially for irregularly shaped or locally difficult-to-define lesions.

Secondly, leveraging the advantages of the Transformer, DETR can handle the detection of multiple lesions simultaneously and capture the features of different lesions at different levels. In CXR images, lesions such as lung nodules and pneumonia may be located in different anatomical regions. DETR's global feature modeling can better perform spatial allocation and relationship modeling, improving the accuracy of multi-lesion detection.

Additionally, unlike YOLO's local feature extraction, DETR can model the global context information in CXR images through self-attention mechanisms, particularly excelling in handling complex backgrounds and multiple overlapping lesions. The self-attention mechanism allows the model to learn relationships between different parts, helping to reduce false positives and false negatives, especially in chest imaging.

However, DETR faces some challenges in CXR image detection. Despite its significant advantages in global feature modeling, DETR performs poorly in detecting small targets, such as tiny lung nodules or early-stage lesions. The Transformer model requires a longer training time to learn the details of small targets, and thus, when the targets are small or inconspicuous, DETR may not be as efficient as detection methods specifically designed for small targets. Additionally, when dealing with complex backgrounds (such as shadows, ribs, and cardiac contours), DETR may overlook some important background information in the image, leading to false positives. This is especially the case when the lesion is very similar to or overlaps with the background, and DETR may fail to distinguish the foreground from the background effectively.

To improve the detection performance of small targets, researchers have made various modifications and optimizations to the standard DETR model. Wu et al. proposed a CDT-CAD [26] model based on DETR. Through comparative experiments on the Vinbig Chest and Chest Det 10 datasets, CDT-CAD demonstrated its effectiveness in identifying chest abnormalities. Müller et al. proposed the Chest X-ray Interpreter (ChEX) [27], which improves recognition performance by enhancing the loss function of DETR.

Table 1 summarizes key prior works in CXR image recognition, highlighting their approaches, strengths, and limitations, which inform the development of CD-DETR.

This manuscript combines Feature Pyramid Networks [28], fusion of contextual information [29], attention mechanisms [30], and optimization of loss functions to propose a DETR optimization model for detecting thoracic diseases in CXR images.

## 3. Methodology

### 3.1 Technology program

The DETR (Detection Transformer) model is an end-to-end object detection model based on the Transformer architecture, which has achieved significant results in the field of object detection. The parallel processing capabilities of DETR allow it to more efficiently utilize hardware resources during training and inference, which helps to accelerate the training process and improve the training efficiency of the model. However, DETR performs poorly in detecting small objects. This is mainly because the features of small objects may be diluted in the global feature map, leading to a decrease in detection accuracy. Therefore, we propose an optimized solution based on the DETR model, as shown in Fig 2, called CD-DETR (Chest Diseases DETR).

- **Selection of backbone.** The CD-DETR model adopts ResNet-50 [31] as the backbone network. ResNet-50 consists of 50 layers of residual blocks, and its residual structure effectively alleviates the problem of vanishing gradients in deep networks, enabling the network to learn more complex feature representations. The architecture of ResNet-50 is highly flexible and is suitable for different input sizes and task requirements. Thanks to its excellent performance, ResNet-50 is widely used in computer vision tasks such as image classification, object detection, and semantic segmentation.

- **Multi-scale feature fusion.** To address the issue of significantly increased computational requirements due to the increased length of the input sequence, the model redesigned the encoder structure and incorporated a multi-scale

**Table 1. Summary of previous works in CXR image recognition.**

| Method | Approach | Strengths | Limitations | Ref |
|---|---|---|---|---|
| Template Matching | Rule-based anomaly detection | Simple, interpretable | Manual feature design, poor scalability | [8] |
| YOLOv5 | CNN-based object detection | Fast, reliable for large targets | Struggles with tiny lesions | [20] [21] |
| DETR | Transformer-based end-to-end detection | Global context modeling | Poor small object detection | [24] |
| CDT-CAD | DETR with context-aware transformers | Effective for chest abnormalities | Complexity increases computation | [26] |

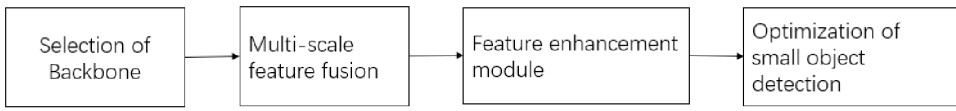

**Fig 2. Model optimization scheme.**

feature fusion strategy [32]. Fusing features at multiple scales not only effectively improves detection accuracy but also avoids the introduction of additional computational costs.

High-Level Feature Processing: The self-attention mechanism is applied to high-level features that contain richer semantic concepts, capturing the relationships between conceptual entities in the image, which helps subsequent modules to perform precise object detection and recognition.

Cross-Level Feature Fusion: By fusing features from multiple levels, the model extracts contextual information around defects, thereby reducing false positives and false negatives caused by the confusion between foreground and background.

- **Feature enhancement module.** The model introduces two feature enhancement mechanisms:

Efficient Channel Attention (ECA-Net): ECA-Net [33] is a channel-based attention mechanism that enhances feature representation through local cross-channel interactions while controlling the increase in computational cost.

Spatial Attention Upsampling (SAU): SAU [34] performs feature upsampling through a spatial attention mechanism, achieving cross-scale feature fusion.

The combination of these modules enables the model to optimize feature representations at different levels. By integrating ECA-Net and SAU modules, the CD-DETR model fuses low-level features extracted by the backbone network with high-level features generated by the Transformer, further capturing global contextual information, thereby improving the overall performance of the model.

- **Optimization for small object detection.** To address the issues of class imbalance and matching instability in small object detection, a dedicated combined Intersection over Union (IoU) loss function, CDIoU, is designed. This loss function reduces the instability caused by minor defects during the matching process and mitigates the impact of class imbalance, thereby improving the detection performance of small objects.

### 3.2 Workflow of the CD-DETR model

The structure of CD-DETR model is seen in Fig 3.

- **Input CXR image.** Let $I \in \mathbb{R}^{H \times W \times C}$ be the input CXR image, where H and W are the height and width of the image, and C is the number of channels.

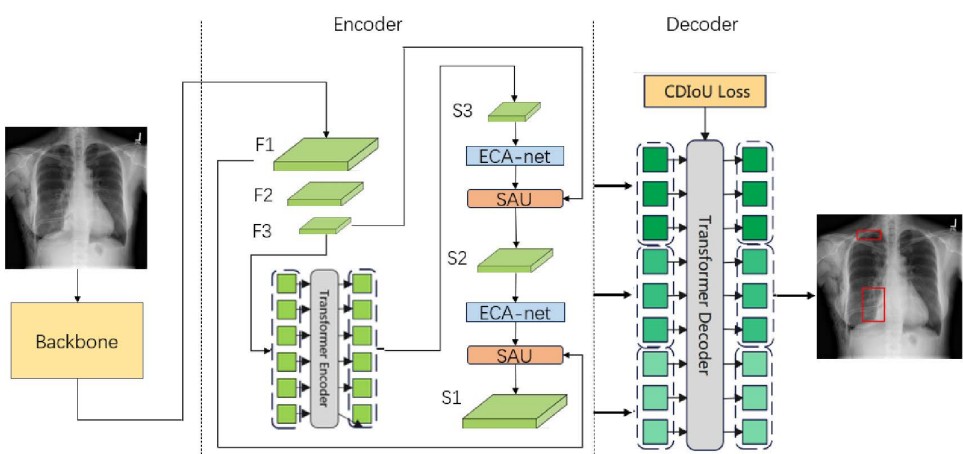

**Fig 3. CD-DETR model structure diagram.**

- **Feature extraction using ResNet-50.** The input image I is passed through the ResNet-50 backbone network to extract feature representations:

$$F_{\text{backbone}} = \text{ResNet} - 50\,(\text{I}) \tag{1}$$

where $F_{\text{backbone}} \in \mathbb{R}^{H' \times W' \times C'}$ is the output feature map, and H′, W′, and C′ represent the height, width, and number of channels of the feature map, respectively.

- **Multi-Scale feature extraction.** From the ResNet-50 output, we extract multiple scales of features $F_{\text{scale}} = \{F_1, F_2, ..., F_n\}$, where each $F_i$ represents features at different scales.
- **Encoder module of the DETR model.** The extracted multi-scale features $F_{scale}$ are passed through the encoder module of the DETR model:

$$F_{encoder} = Encoder\,(F_{scale}) \tag{2}$$

where $F_{encoder} \in \mathbb{R}^{N \times D}$ is the encoded feature matrix, with N representing the number of tokens or positions, and D being the dimensionality of each feature vector.

- **Enhancing feature representation using ECA-Net.** The features $F_{encoder}$ obtained from the encoder module are processed through the ECA-Net self-attention mechanism to enhance the feature representation by enabling local cross-channel interactions:

$$F_{\text{attention}} = \text{ECA} - \text{Net}\,(F_{encoder}) \tag{3}$$

where $F_{\text{attention}} \in \mathbb{R}^{N \times D}$ is the output of the self-attention module.

- **Feature fusion.** The features from the ResNet-50 backbone $F_{\text{backbone}}$ and the features from the encoder module $F_{\text{attention}}$ are fused by SAU module to capture contextual information around defects and reduce the confusion between foreground and background:

$$F_{\text{fusion}} = \text{SAU}\,(F_{\text{backbone}}, F_{\text{attention}}) \tag{4}$$

where $F_{\text{fusion}} \in \mathbb{R}^{N \times D}$ represents the fused features containing both low-level and high-level information.

- **CDIoU loss for small object detection.** In the decoder module, a specialized joint cross-IoU loss function (CDIoU) is applied to address class imbalance and instability during matching in small target detection:

$$\mathcal{L}_{CDIoU} = CDIoULoss\,(y_{pred}, y_{true}) \tag{5}$$

where $y_{pred}$ represents the predicted target locations and classes, and $y_{true}$ represents the ground truth target locations and classes.

The CDIoU loss for small disease, which is based on EIoU (Efficient-IoU) Loss [35]. EIoU is an advanced loss function introduced to address the challenges in traditional IoU-based loss functions, especially in tasks like object detection where high precision and accuracy are essential. Efficient IoU loss is designed to improve the performance of models, particularly for small object detection, by providing more accurate optimization and better handling of object overlap and spatial localization. Our specially designed CDIoU Loss for small disease leverages classification scores and optimizes all components in a unified direction. We've introduced Defect IoU Loss to accommodate this enhancement, as shown in Equation 6:

$$L_{\text{CDIoU}} = 1 + rIoU - IoU + \frac{\rho^2 o\left(o, o^{gt}\right)}{(w^c)^2 + (h^c)^2} + \frac{\rho^2\left(w, w^{gt}\right)}{(w^c)^2} + \frac{\rho^2\left(h, h^{gt}\right)}{(h^c)^2} \tag{6}$$

where $o$ and $o^{gt}$ represent the centers of the prediction box and ground truth box, respectively, $w^c$ and $h^c$ are the width and height of the smallest enclosing box of the two boxes, and $\rho^2\,(\,)$ denotes the Euclidean distance between two points. $r$ represents the classification scores.

- **Decoder output – object detection results.** The decoder module outputs the final object detection results youty_ {\text{out}}, which include the class, location, and confidence of each detected object:

$$y_{out} = Decoder\,(F_{\text{fusion}})$$ (7)

where $y_{out}$ consists of the detected objects' class labels, bounding box coordinates, and confidence scores.

Through these steps, the CD-DETR model utilizes techniques such as multi-scale feature fusion, self-attention mechanism, and specialized loss function to optimize the performance of object detection tasks, especially in small object detection.

## 4. Experiment

### 4.1 Dataset

- **NIH Chest X-ray dataset.** The NIH Chest X-ray dataset [36] is a large publicly available dataset for chest X-ray (CXR) image analysis, primarily used in the research of deep learning models for medical image classification and disease detection. It was released by the National Institutes of Health (NIH) and is designed to support the development and evaluation of algorithms that can assist in diagnosing diseases from CXR images. The dataset contains over 112,000 frontal-view chest X-ray images from 30,805 unique patients. The images are labeled with 14 different diseases including pneumonia, tuberculosis, lung cancer, and other common chest conditions.

The NIH Chest X-ray dataset is one of the largest publicly available datasets for CXR image classification. It covers a wide variety of chest conditions and offers a diverse set of images from different patients, which helps to ensure that the trained models can generalize well to unseen data. The NIH dataset offers multi-label annotations, which means each image can be associated with multiple diseases simultaneously, allowing the model to learn to identify multiple conditions from a single image. The NIH Chest X-ray dataset has become a benchmark dataset for testing the performance of various deep learning algorithms in the medical imaging domain.

- **VinBigData dataset.** Provided by Vingroup Big Data Institute, this dataset contains 18000 scans annotated by experienced radiologists. The annotations were collected through VinBigData's web-based platform VinLab [37]. The VinBigData dataset is a large-scale, high-quality dataset designed specifically for training and evaluating deep learning models for CXR image recognition. The dataset contains high-resolution chest X-ray images, which are essential for the development of accurate image recognition models capable of detecting subtle abnormalities. The VinBigData dataset is an excellent choice for CXR image recognition and disease detection model research due to its large scale, diverse disease categories, multi-label annotations, and high-quality images.

### 4.2 Training setup

The CD-DETR model was trained using the PyTorch framework on an NVIDIA A100 GPU with 40GB memory. In the experiment, we utilized a Resnet model [31] pre-trained on ImageNet as the backbone network for extracting multi-scale features. The training configuration is detailed in Table 2 below to ensure reproducibility.

### 4.3 Evaluation indicators

Precision and Recall are two fundamental evaluation metrics in classification problems, particularly useful for assessing the performance of models in situations where the classes are imbalanced or where the cost of false positives and false negatives is uneven.

**Table 2. Training parameters for CD-DETR model.**

| Parameter | Value |
|---|---|
| Backbone Network | ResNet-50 |
| Optimizer | AdamW |
| Learning Rate | 0.0002 |
| Weight decay | 0.0001 |
| Batch Size | 16 |
| Epochs | 72 |
| Number of Encoder Layers | 6 |
| Number of Decoder Layers | 6 |

Precision measures the proportion of true positive predictions (correctly identified instances of the positive class) out of all the instances that were predicted as positive by the model.

$$Precision = \frac{\| TP \|}{\| TP \| + \| FP \|}$$

(8)

Recall measures the proportion of true positive predictions (correctly identified instances of the positive class) out of all the actual instances of the positive class (i.e., the total actual positives).

$$Recall = \frac{\| TP \|}{\| TP \| + \| FN \|}$$

(9)

Recall is particularly important in scenarios where the consequences of missing a positive case (false negative) are severe. For instance, in medical screenings, failing to identify a patient with a disease (false negative) could have dire consequences. Recall should be prioritized when the cost of false negatives is high. We would prefer to detect as many cases as possible (even if some false positives occur) to avoid missing critical diagnoses.

## 4.4 Experimental result

On the NIH Chest X-ray Dataset, we compared the CD-DETR with other advanced DETR variants and CNN-based models. Table 3 displays the results of these comparisons. It is evident from the table that our model achieved the highest precision and recall with fewer parameters.

Our method outperforms high-performance models such as Deformable-DETR, Group-DETR, and Focus-DETR, achieving an average 5% increase in detection accuracy. These models face challenges in identifying diseases with weak features or in complex backgrounds due to their approach of merging multi-scale features into a single input for the encoder, which not only increases computational costs but also risks feature confusion. In contrast, our model integrates a feature fusion process between the encoder and decoder, ensuring that small disease-related features in high-resolution maps are not overlooked during training.

Compared to the baseline model RT-DETR, our model, CD-DETR, achieves more than a 2% improvement in precision. This enhancement is due to the fact that RT-DETR's interpolation upsampling and convolutional pooling fail to preserve the semantic consistency of features. Additionally, RT-DETR's sparse sampling approach for object query selection limits the decoder's performance. In contrast, the SAU operator introduced in our model ensures the consistency of upsampled features, making it well-suited for self-attention-based networks.

To assess overfitting, we split the NIH Chest X-ray dataset into 80% training, 10% validation, and 10% test sets, and the VinBigData sample similarly. Training and validation losses were monitored over 72 epochs, with convergence

**Table 3. Comparison of different methods in NIH Chest X-ray dataset dataset.**

| Model | Backbone | #Epcohs | #Param(M) | Precision (%) | Recall (%) |
|---|---|---|---|---|---|
| Faster-RCNN [22] | R50 | 24 | – | 78.3 | 65.2 |
| YOLOv5 [38] | – | 275 | – | 80.2 | 67.7 |
| YOLOv7 [39] | – | 275 | – | 82.7 | 80.1 |
| IYOLOv7 [40] | – | 275 | – | 87.2 | 83.5 |
| YOLOv8 [41] | – | 275 | | 87.9 | 84.2 |
| Deformable-DETR [42] | R50 | 50 | 40.2 | 81.6 | 78.5 |
| Conditional-DETR [43] | R50 | 50 | 43.3 | 82.4 | 68.3 |
| Group-DETR [44] | R50 | 50 | 44.1 | 78.6 | 75.6 |
| Focus-DETR [45] | R50 | 36 | 49.5 | 85.5 | 82.2 |
| RT-DETR [46] | R50 | 72 | 41.4 | 86.1 | 83.5 |
| CD-DETR (Ours) | **R50** | **72** | **39.3** | **88.3** | **86.6** |

Note: The missing part of the data is based on CNN methods, which are not comparable to those based on DETR methods.

observed around epoch 60, indicating no significant divergence (validation loss stabilized at ~ 0.15 vs. training loss ~0.12). Techniques such as random flipping and rotation for data augmentation, combined with a weight decay of 0.0001 in the AdamW optimizer, further prevented overfitting. Test set performance (88.3% precision, 86.6% recall on NIH) closely aligned with validation results, confirming the model's generalization ability.

## 4.5 Generalization study

To assess the robustness and wide applicability of our model for detecting chest diseases, we performed experiments on a sample from the VinBigData dataset. The outcomes, presented in Table 4, highlight the model's outstanding detection performance. When compared with the top-performing algorithm, Hybrid-YOLO, our approach achieves comparable precision while enhancing the recall rate by 2.3%. This indicates our method's ability to identify as many diseases as possible, benefiting from the innovative aspects we proposed, which are more aligned with actual medical applications. Additionally, our method shows significant improvements compared to DETR-based methods, demonstrating strong robustness and the ability to achieve better results across various data distributions.

## 5. Discussion

### 5.1 Theoretical and practical contributions

Significant progress has been made in detecting chest diseases through the improved DETR model, especially in detecting minor pathologies. This advancement has substantial practical significance for the development of medical systems.

**Table 4. Comparison of different methods in VinBigData dataset.**

| Model | Precision (%) | Recall (%) |
|---|---|---|
| YOLO-5 [38] | 75.2 | 71.3 |
| Hybrid-YOLO [47] | 85.4 | 81.2 |
| Deform[able-DETR [42] | 78.2 | 74.7 |
| Focus-DETR [45] | 80.5 | 75.6 |
| RT-DETR [46] | 81.3 | 77.2 |
| CD-DETR(Ours) | **85.2** | **83.5** |

1. Enhancing Detection Efficiency and Accuracy: The optimized detection model employing deep learning technology can significantly improve the efficiency and accuracy of identifying chest diseases from chest X-ray images. This aids in the timely discovery of chest diseases, enhances the diagnostic efficiency of doctors, and is particularly beneficial in reading centers that need to process a large volume of medical images promptly.

2. Reducing Costs and Risks: There is a severe shortage of radiologists in the western regions of China. In rural and township hospitals in the west, although radiological medical equipment is basically widespread, there is a lack of professionally capable doctors. Patients often have to wait for several days to receive diagnostic results after completing imaging examinations, which may pose the risk of delayed treatment. Compared to traditional medical models, the AI-assisted centralized reading model reduces manpower requirements, lowers examination costs, and minimizes the risk of patients' delayed treatment.

3. High Recall Rate of CD-DETR Model: Compared to other advanced models, the CD-DETR model has a higher recall rate, which is significant for screening in reading centers. AI screening should aim to identify as many diseased CXRs as possible for doctors to complete the diagnosis. Even if some CXRs identified by the model do not have chest diseases, because a doctor will ultimately confirm, there will be no misdiagnosis. If the CXR contains disease information, but the model fails to identify, it could increase the risk of missed diagnoses.

Additionally, AI-assisted reading enhances the accessibility of medical services, providing remote diagnostic services for remote areas or regions with scarce medical resources, increasing the opportunity for local residents to access high-quality medical services. In the event of public health crises, such as pandemic outbreaks, AI-assisted reading can quickly process a large number of cases, providing support for epidemic control. AI-assisted centralized reading not only improves the quality and efficiency of medical services but also has a positive impact on a broader social level, promoting equality and progress in the field of healthcare.

## 5.2 Limitations and prospects

The current study primarily utilized the NIH Chest X-ray dataset and a subset of the VinBigData dataset. While these datasets are extensive and diverse, they may not fully represent the global variability in CXR images, particularly in terms of different patient populations, imaging equipment, and disease prevalence. This could limit the generalizability of the CD-DETR model to other regions or specific patient groups. Future research should expand the dataset to include more diverse and geographically distributed CXR images. This will help in validating the model's performance across different healthcare settings and patient demographics, ensuring broader applicability.

The model may still struggle with CXR images that have complex backgrounds, such as those with multiple overlapping structures or artifacts. This can lead to false positives and false negatives, particularly for small and subtle lesions. Further research into advanced attention mechanisms and context-aware feature extraction techniques can help the model better distinguish between relevant and irrelevant information in complex images. This will improve the robustness of the model in handling a wider range of CXR image qualities and conditions. In recent studies, the pACP-HybDeep [48] model has optimized the structure using a transformer based on binary tree growth and a deep hybrid learning approach. Target-CLP [49] and DeepAIPs-Pred [50] have enhanced recognition accuracy through a weighted feature integration method. AIPs-DeepEnC-GA [51] proposed a deep learning method based on an intelligent genetic algorithm for the prediction of anti-inflammatory peptides, achieving significant progress. These methods have many technical advantages worth learning from. In future research, we will integrate the strengths of recent studies to further improve the accuracy of the model.

Additionally, developing a seamless integration of the CD-DETR model into existing clinical workflows can significantly enhance its adoption and impact. This includes creating user-friendly interfaces for radiologists and clinicians, and ensuring interoperability with hospital information systems (HIS) and picture archiving and communication systems (PACS).

Collaborating with healthcare providers and IT specialists to design and implement a clinical decision support system (CDSS) that incorporates the CD-DETR model. This system can provide real-time diagnostic suggestions and alerts, improving the efficiency and accuracy of radiological reporting.

In summary, future directions for this study include:

- Real-Time Deployment: Optimize CD-DETR for edge devices using techniques like model pruning or quantization to enable real-time inference in low-resource settings.

- Global Dataset Validation: Expand evaluation to include CXR datasets from diverse regions (e.g., Africa, South America) to ensure robustness across patient populations and imaging conditions.

- Clinical Integration: Develop a clinical decision support system (CDSS) incorporating CD-DETR, with user-friendly interfaces and compatibility with hospital systems (e.g., PACS), to enhance adoption in centralized reading workflows.

- Advanced Feature Extraction: Explore novel attention mechanisms or background suppression techniques to further improve detection in complex CXR images with overlapping structures.

## 6. Conclusion

The CD-DETR model has demonstrated significant advancements in the detection of thoracic diseases from CXR images, especially in detecting minor diseases that are often challenging to identify. The model's innovative integration of ECA-Net and SAU for feature enhancement, along with the CDIoU loss function for optimized small target detection, has resulted in improved precision and recall rates. This study not only contributes theoretically by providing a robust model for CXR analysis but also practically by offering a solution to address the radiologist shortage in underdeveloped regions. The AI-assisted centralized reading method facilitated by CD-DETR has the potential to enhance diagnostic efficiency, reduce costs, and improve patient outcomes. Future work will focus on expanding the dataset, optimizing algorithms, and ensuring the model's real-time performance to further refine the detection capabilities and broaden its applicability in various medical imaging scenarios. The CD-DETR model paves the way for more accurate and effective detection of thoracic diseases, contributing to the evolution of object detection techniques and advancing healthcare equality and progress.

### Author contributions

**Conceptualization:** Shasha Yu.

**Data curation:** Shasha Yu, Peng Zhou.

**Formal analysis:** Shasha Yu, Peng Zhou.

**Methodology:** Peng Zhou.

**Software:** Peng Zhou.

**Validation:** Peng Zhou.

**Writing – original draft:** Peng Zhou.

**Writing – review & editing:** Shasha Yu.

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
