## [Decision Letter · Decision Letter 0]

20 Feb 2025

PONE-D-25-02642An Optimized Transformer Model for Efficient Detection of Thoracic Diseases in Chest X-rays with Multi-Scale Feature FusionPLOS ONE

Dear Dr. Zhou,

Thank you for submitting your manuscript to PLOS ONE. After careful consideration, we feel that it has merit but does not fully meet PLOS ONE’s publication criteria as it currently stands. Therefore, we invite you to submit a revised version of the manuscript that addresses the points raised during the review process.

We look forward to receiving your revised manuscript.

Kind regards,

Asadullah Shaikh, Ph.D.

Academic Editor

PLOS ONE

[6330].

Reviewers' comments:

Reviewer's Responses to Questions

**Comments to the Author**

1. Is the manuscript technically sound, and do the data support the conclusions?

Reviewer #1: Partly

Reviewer #2: Yes

2. Has the statistical analysis been performed appropriately and rigorously? 

Reviewer #1: Yes

Reviewer #2: Yes

3. Have the authors made all data underlying the findings in their manuscript fully available?

Reviewer #1: Yes

Reviewer #2: Yes

4. Is the manuscript presented in an intelligible fashion and written in standard English?

Reviewer #1: Yes

Reviewer #2: Yes

5. Review Comments to the Author

Reviewer #1: 1. In the abstract, the authors should mention how much improvement the proposed model achieved compared to existing methods.

2. At the end of the introduction section, the authors should mention the main contributions of the proposed model in points.

3. In the related work, the authors need to cite all the text with a supporting article properly.

4. The authors need to provide all the parameters used for model training in the form of a table.

5. The authors need to compare the results of the proposed model with the existing state-of-the-art model to validate its effectiveness.

6. The authors need to cite the recent deep learning-based predictors such as pACP-HybDeep, TargetCLP, DeepAIPs-Pred, and AIPs-DeepEnC-GA to provide an updated overview to the readers

7. How the authors evaluate the overfitting of the proposed model.

8. What should be the future directions of the proposed study?

Reviewer #2: 1. Check for grammatical, spelling, and punctuation errors.

2. Use more significant keywords.

3. Write the result clearly in the abstract.

4. There are too many paragraphs in the introduction. Maximum 5-6 paragraphs are good.

5. Write the novelty/contribution of this work clearly in the introduction section.

6. Use a summary table for previous works in the related works.

7. Cite the below publications in the paper:

>> https://doi.org/10.1016/j.imu.2021.100741

>> https://doi.org/10.1109/IICAIET55139.2022.9936837

>> https://doi.org/10.1109/ICCCNT56998.2023.10307223

8. Improve the figure quality.

6. PLOS authors have the option to publish the peer review history of their article (what does this mean? ). If published, this will include your full peer review and any attached files.

**Do you want your identity to be public for this peer review?** For information about this choice, including consent withdrawal, please see our Privacy Policy .

Reviewer #1: No

Reviewer #2: No

---

## [Decision Letter · Decision Letter 1]

6 Apr 2025

An Optimized Transformer Model for Efficient Detection of Thoracic Diseases in Chest X-rays with Multi-Scale Feature Fusion

PONE-D-25-02642R1

Dear Dr. Zhou,

We’re pleased to inform you that your manuscript has been judged scientifically suitable for publication and will be formally accepted for publication once it meets all outstanding technical requirements.

Kind regards,

Asadullah Shaikh, Ph.D.

Academic Editor

PLOS ONE

Additional Editor Comments (optional):

Reviewers' comments:

Reviewer's Responses to Questions

**Comments to the Author**

1. If the authors have adequately addressed your comments raised in a previous round of review and you feel that this manuscript is now acceptable for publication, you may indicate that here to bypass the “Comments to the Author” section, enter your conflict of interest statement in the “Confidential to Editor” section, and submit your "Accept" recommendation.

Reviewer #1: All comments have been addressed

Reviewer #2: All comments have been addressed

2. Is the manuscript technically sound, and do the data support the conclusions?

Reviewer #1: Yes

Reviewer #2: Yes

3. Has the statistical analysis been performed appropriately and rigorously? 

Reviewer #1: Yes

Reviewer #2: Yes

4. Have the authors made all data underlying the findings in their manuscript fully available?

Reviewer #1: Yes

Reviewer #2: Yes

5. Is the manuscript presented in an intelligible fashion and written in standard English?

Reviewer #1: Yes

Reviewer #2: Yes

6. Review Comments to the Author

Reviewer #1: The required comments are successfully incorporated and paper is significantly improved. The paper can be accepted from my side

Reviewer #2: Thanks to the authors for addressing all comments and revising the manuscript. The quality of the manuscript has been improved.

7. PLOS authors have the option to publish the peer review history of their article (what does this mean? ). If published, this will include your full peer review and any attached files.

**Do you want your identity to be public for this peer review?** For information about this choice, including consent withdrawal, please see our Privacy Policy .

Reviewer #1: No

Reviewer #2: No

---

## [Editor Report · Acceptance letter]

PONE-D-25-02642R1

PLOS ONE

Dear Dr. Zhou,

I'm pleased to inform you that your manuscript has been deemed suitable for publication in PLOS ONE. Congratulations! Your manuscript is now being handed over to our production team.

Kind regards,

on behalf of

Prof. Asadullah Shaikh

Academic Editor

PLOS ONE